# Emergence of collective oscillations in adaptive cells

Shou-Wen Wang ⓘ [1,2,3]* & Lei-Han Tang ⓘ [1,4,5]*

Collective oscillations of cells in a population appear under diverse biological contexts. Here, we establish a set of common principles by categorising the response of individual cells against a time-varying signal. A positive intracellular signal relay of sufficient gain from participating cells is required to sustain the oscillations, together with phase matching. The two conditions yield quantitative predictions for the onset cell density and frequency in terms of measured single-cell and signal response functions. Through mathematical constructions, we show that cells that adapt to a constant stimulus fulfil the phase requirement by developing a leading phase in an active frequency window that enables cell-to-signal energy flow. Analysis of dynamical quorum sensing in several cellular systems with increasing biological complexity reaffirms the pivotal role of adaptation in powering oscillations in an otherwise dissipative cell-to-cell communication channel. The physical conditions identified also apply to synthetic oscillatory systems.

[1] Beijing Computational Science Research Center, Beijing 100094, China. [2] Department of Engineering Physics, Tsinghua University, Beijing 100086, China. [3] Department of Systems Biology, Harvard Medical School, Boston, MA 02115, USA. [4] Department of Physics and Institute of Computational and Theoretical Studies, Hong Kong Baptist University, Hong Kong, China. [5] State Key Laboratory of Environmental and Biological Analysis, Hong Kong Baptist University, Hong Kong, China. *email: shouwen_wang@hms.harvard.edu; lhtang@csrc.ac.cn

Homogeneous cell populations are able to exhibit a rich variety of organised behaviour, among them periodic oscillations. During mound formation of starved social amoebae, cyclic AMP waves guide migrating cells towards the high density region[1–5]. Elongation of the vertebrate body axis proceeds with a segmentation clock[6,7]. Multicellular pulsation has also been observed in nerve tissues[8], during dorsal closure in late stage drosophila embryogenesis[9–12], and more[13]. In these examples, communication through chemical or mechanical signals is essential to activate quiescent cells. Dubbed dynamical quorum sensing (DQS) to emphasise the role of increased cell density in triggering the auto-induced oscillations, this class of behaviour lies outside the well-known Kuramoto paradigm of oscillator synchronisation[14,15].

Interestingly, auto-induced oscillations have also been reported in situations without an apparent biological function. A case in point is otoacoustic emission (OAE), where a healthy human ear emits sound spontaneously in a silent environment[16,17]. Anatomically, sound is generated by hair bundles, the sensory units of hair cells that detect sound with ultra-high sensitivity[18–20]. Another example is glycolytic oscillations of yeast cells which can be induced across different laboratory conditions[21–26]. This type of phenotypic behaviour may not confer benefits to the organism, so their existence is puzzling.

Here, we consider a population of cells attempting to modulate temporal variations of the extracellular concentration of a protein or analyte, or a physical property of their environment, by responding to it. The response of a cell to the external property, or signal, can be mediated by an arbitrary intracellular biochemical network. By focusing on the frequency-resolved cellular response, we report a generic condition for collective oscillations to emerge, and show that it is satisfied when cells affect the signal in a way that adapts to slow environmental variations, i.e., cells respond to signal variation rather than to its absolute level. In particular, we prove the existence of an active frequency regime, where adaptive cells anticipate signal variation and attempt to amplify the signal. Sustained collective oscillations emerge when a cell population, beyond a critical density, communicates spontaneously through such a channel.

We provide a physical explanation of oscillations in terms of energy-driven processes, with adaptive cells outputting energy in the active frequency regime upon stimulation. For mechanical signals, the energy output is directly observable as work on the environment. For chemical signals, chemical free energy is transferred during the release of molecules into the extracellular medium. Together with the measurable response of individual cells, quantitative predictions of the oscillation frequency and its dependence on cell density become possible.

The adaptive cellular response highlighted in this work is shown to underlie several known examples of DQS, and possibly glycolytic oscillations in yeast cell suspensions. The ubiquity of adaptation[27–37] in biology may also explain the emergence of inadvertent oscillations. We discuss implications and predictions of this general mechanism at the end of the paper, in connection with previous experimental and modelling work.

## Results

**Necessary conditions for auto-induced oscillations.** We begin by considering a scenario of mechanical oscillations, as illustrated in Fig. 1a. Later we will show that the same results hold for chemical oscillations. The cells are spatially close enough so that they could be regarded as under the same environment. Here, the extracellular signal $s$ is taken to be the deformation of the mechanical environment, which is both sensed and modified by participating cells. The cell activity that affects the environment

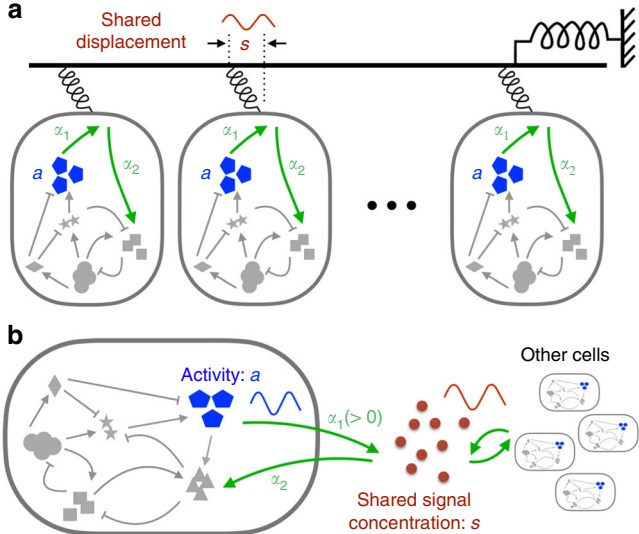

**Fig. 1 Spontaneous oscillations in a communicating cell population. a** The scenario of mechanical oscillations where cells communicate via a shared displacement $s$ of the physical environment. Activity $a$ of a cell against the displacement is regulated by a hidden intracellular network which responds to $s$ through a mechanical sensor. **b** Illustration of chemical oscillations where cells interact via a shared extracellular signal $s$. The signal is sensed and secreted by individual cells.

is denoted by a variable $a$. Its dynamics is controlled by an unspecified intracellular regulatory network that responds to $s$ through a mechanical sensor. To see how the intracellular activities might disrupt stasis in an equilibrium state of $s$, we consider the following Langevin equation:

$$\gamma \dot{s} = F(s) + \sum_{j=1}^{N} \alpha_1 a_j + \xi. \tag{1}$$

Here $\gamma$ is the friction coefficient, $F(s)$ the external force that tries to restore the physical environment, $\xi$ the thermal noise, and the sum represents the total force created by $N$ active cells in a unit volume, whose strength is set by $\alpha_1 > 0$. In general, the cell activity depends on the past history of the signal. Upon a small change of $s$, the average response of the activity of $j$th cell satisfies

$$\langle a_j(t) \rangle = \langle a_j \rangle_u + \int_{-\infty}^{t} R_{a_j}(t - \tau) \langle s(\tau) \rangle d\tau, \tag{2}$$

with $\langle \cdot \rangle$ and $\langle \cdot \rangle_u$ denoting noise average with and without an external time-varying signal, respectively. Without loss of generality, we set the stationary activity $\langle a_j \rangle_u$ to zero. The activity response function $R_a$ is a property of the intracellular molecular network, which can be computed for specific models[38,39] or measured directly in single-cell experiments[3,4,19,40]. In general, $R_a$ may depend on the ambient signal level $s$ of the cell.

The shared signal $s$ offers a means to synchronise the activities of cells. We derive here a matching condition for $s$ and the $a$'s to enter a positive signal relay. Expressing Eq. (2) in Fourier form, we have $\langle \tilde{a}_j(\omega) \rangle = \tilde{R}_{a_j}(\omega) \langle \tilde{s}(\omega) \rangle$. For weak disturbances, the restoring force in Eq. (1) can be approximated by a linear one, i.e., $F(s) \simeq -Ks$. Consequently, $\langle \tilde{s}(\omega) \rangle = \sum_{j=1}^{N} \alpha_1 \tilde{R}_s(\omega) \langle \tilde{a}_j(\omega) \rangle$, where

$$\tilde{R}_s = \frac{1}{K - i\gamma\omega} \tag{3}$$

is the signal response function, with $i$ the imaginary unit. For identical cells, these equations yield an oscillatory solution $\tilde{a}(\omega_o) \neq 0$ provided $N\alpha_1 \tilde{R}_a(\omega_o)\tilde{R}_s(\omega_o) = 1$. To gain more insight,

we express the two response functions in their amplitudes and phase shifts, i.e. $\tilde{R}_a \equiv |\tilde{R}_a| \exp(-i\phi_a)$ and $\tilde{R}_s \equiv |\tilde{R}_s| \exp(-i\phi_s)$. Then, the cell density $N = N_o$ and the selected frequency $\omega_o$ at the onset of collective oscillations are determined by,

$$\phi_a(\omega_o) = -\phi_s(\omega_o), \qquad (4)$$

$$|\tilde{R}_a(\omega_o)\tilde{R}_s(\omega_o)| = (\alpha_1 N_o)^{-1}. \qquad (5)$$

These are essentially conditions of linear instability for the quiescent state expressed in terms of the single-cell and signal response functions, and constitute our first main result. For inhomogeneous cell populations, one simply replaces $R_a$ by its population average $\overline{R}_a \equiv N^{-1}\sum_{j=1}^{N} R_{a_j}$.

Under the assumption of additive signal release from individual cells as expressed by Eq. (1), we now have a mathematical prediction for the onset density $N_o$ and oscillation frequency $\omega_o$. Let $\alpha_2 \sim |\tilde{R}_a|$ be the sensitivity of the cell activity against $s$. We introduce a signal relay efficiency $\overline{N} \equiv N\alpha_1\alpha_2$, which also sets the coupling strength of cellular activities through the signal. Oscillations start at the critical coupling strength $\overline{N}_o = N_o\alpha_1\alpha_2$. Eq. (5) simply states that, at the selected frequency $\omega_o$, signal amplification through the collective action of $N_o$ cells compensates signal loss from dissipative forces acting on $s$, e.g., friction for a mechanical signal or degradation/dilution for a chemical signal. The frequency $\omega_o$ is chosen such that phase shifts incurred in the forward and reverse medium-cell transmissions match each other (Eq. (4)).

Although we mainly focus on the emergence of collective oscillations as cell density increases, oscillation death at high cell densities through a continuous transition, as observed in certain experimental systems, may also fulfil the self-consistency condition (Eqs. (4–5)). An example is provided in Supplementary Note 6 (see also Supplementary Fig. 15), where, due to the nonlinear properties of the intracellular circuit, the cell activity becomes less responsive at elevated signal intensity. To make use of our procedure, the cell-density dependence of the linear response functions needs to be considered.

**Cell-to-signal energy flow.** Auto-induced collective oscillations must be driven by intracellular active processes. These active components of the system give a nonequilibrium character to the activity response[41–44] and furthermore enable energy flow from the cell to the signal upon periodic stimulation, an interesting physical phenomenon left unnoticed so far.

To set the stage, we turn to basic considerations of non-equilibrium thermodynamics[45,46]. The shared signal $s$ as illustrated in Fig. 1 typically follows a dissipative dynamics such as Eq. (1). When the medium is close to thermal equilibrium, the Fluctuation-Dissipation Theorem (FDT) relates the imaginary component $\tilde{R}_s''$ of the signal response $\tilde{R}_s$ to its spontaneous fluctuation $\tilde{C}_s$ induced by thermal noise[38,39,47]: $2T\tilde{R}_s''(\omega) = \omega\tilde{C}_s(\omega)$, where $\tilde{C}_s(\omega) = \langle|\tilde{s}(\omega)|^2\rangle_u$ is the spectral amplitude of the signal, and $T$ is the temperature. This relation demands $\tilde{R}_s''(\omega)$ to be positive at all frequencies. Hence, the dissipative nature of the physical environment translates into a phase delay, i.e., $\phi_s \equiv -\arg(\tilde{R}_s) \in (-\pi, 0)$. Under the over-damped signal dynamics (Eq. (1)), Eq. (3) gives

$$\phi_s(\omega) \equiv -\arg(\tilde{R}_s(\omega)) = -\tan^{-1}(\omega\tau_s) \in \left(-\frac{\pi}{2}, 0\right), \qquad (6)$$

where $\tau_s = \gamma/K$ is the signal relaxation time. (The situation $-\pi < \phi_s(\omega) < -\pi/2$ occurs at high frequencies when the dynamics of $s$ is underdamped.) On the other hand, a leading phase as required by Eq. (4) for the intracellular signal relay violates the FDT. In the present case, active cells play the role of the

out-of-equilibrium partner. We have calculated the work done by one of the cells on the signal when the latter oscillates at a frequency $\omega$ (see Supplementary Note 1). The output power $\dot{W} \equiv \langle \dot{s} \cdot \alpha_1 a \rangle$, i.e., the averaged value of the product between signal velocity ($\dot{s}$) and force from an individual cell ($\alpha_1 a$), is given by

$$\dot{W} \simeq -\alpha_1\omega\tilde{R}_a''(\omega)\langle|\tilde{s}(\omega)|^2\rangle$$
$$= \alpha_1\omega|\tilde{R}_a(\omega)|\sin\phi_a(\omega)\langle|\tilde{s}(\omega)|^2\rangle. \qquad (7)$$

The energy flux is positive, i.e., flowing from the cell to the signal, when $a$ has a phase lead over $s$, re-affirming Eq. (4) as a necessary condition on thermodynamic grounds. Stimulated energy release from an active cell to the signal as expressed by Eq. (7) constitutes our second main result in this paper.

Eq. (7) can also be used to calculate the energy flux for an arbitrary signal time series $s(t)$, provided the linear response formula Eq. (2) applies. In particular, thermal fluctuations of $s$ in the quiescent state may activate a net cell-to-signal energy flow. The total power is obtained by integrating contributions from all frequencies. Previous experiments from Hudspeth lab yielded a phase-leading response of hair bundles to mechanical stimulation at low frequencies[19]. The same group also showed that energy can be extracted from the hair bundle via a slowly oscillating stimulus[18].

**Chemical oscillations.** The criteria given by Eqs. (4–5) apply equally to chemical oscillations illustrated in Fig. 1b. In contrast to the mechanical system, Eq. (1) at $\gamma = 1$ becomes a rate equation for the extracellular concentration $s$ of the signalling molecules. The term $F(s)$ (negative) gives the degradation or dilution rate of $s$ in the medium, while individual cells secrete the molecules at a rate proportional to their activity $a$. As the signalling molecules are constantly produced and degraded, chemical equilibrium is often violated even in the steady state. Nevertheless, $F(s)$ usually plays the role of a stabilising force so that the signal response function $\tilde{R}_s(\omega)$ has the same phase-lag behaviour as the mechanical case. Release of the molecules by the communicating cells must be phase-leading so as to drive oscillatory signalling.

**Adaptive cells show phase-leading response.** Apart from the aforementioned hair bundles, phase-leading response to a low frequency signal has also been reported in the activity of *E. coli* chemoreceptors[40] and in the osmo-response in yeast[36]. Interestingly, all three of these cases are examples of adaptive sensory systems whose response to a step signal at $t = 0$ is shown in Fig. 2a. The small activity shift $\epsilon$ at long times is known as the adaptation error. Figure 2b shows the response of the same system under a sinusoidal signal. The low frequency response exhibits a phase lead while the high frequency one has a phase lag. Below, we show that the sign switch in the phase shift of an adaptive variable is an inevitable consequence of causality.

From the causality condition $R_a(t < 0) = 0$, the real ($\tilde{R}_a'$) and imaginary ($\tilde{R}_a''$) part of the response function in frequency space satisfy the Kramers-Krönig relation[48]:

$$\tilde{R}_a'(\omega) = \frac{2}{\pi}\int_0^\infty \tilde{R}_a''(\omega_1)\frac{\omega_1}{\omega_1^2 - \omega^2}d\omega_1. \qquad (8)$$

For a step signal of unit strength, Eq. (2) yields

$$\epsilon = \langle a(\infty)\rangle - \langle a\rangle_u = \int_0^\infty R_a(\tau)d\tau = \lim_{\omega\to 0}\tilde{R}_a'(\omega). \qquad (9)$$

Comparing Eqs. (8) and (9) in the limit $\omega \to 0$ and assuming $\epsilon$ to be sufficiently small, we see that $\tilde{R}_a''(\omega)$ inside the integral must

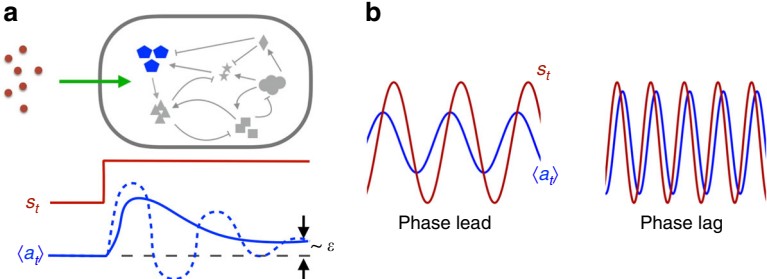

**Fig. 2 Dynamical response of an intracellular adaptive variable $a$. a** Response to a stepwise signal: after a transient response, $a$ returns to its pre-stimulus state (within a small error $\epsilon$). In the simplest case, the transient response is controlled by the activity shift timescale $\tau_a$ and the circuit feedback timescale $\tau_y$. Solid and dashed lines correspond to over-damped ($\tau_a \ll \tau_y$) and under-damped ($\tau_a \gg \tau_y$) situations, respectively. **b** Response to a sinusoidal signal at low (left) and high (right) frequencies. The phase shift $\phi_a$ switches sign.

change sign. In other words, both phase-leading ($\tilde{R}_a'' < 0$) and lagged ($\tilde{R}_a'' > 0$) behaviour are present across the frequency domain. This is our third result.

Adaptation plays a key role in biochemical networks[27,28], and especially in sensory systems[20,29,31–36]. Connection between adaptation and collective oscillations has been implicated in previous works[4,49,50]. With the mathematical results presented above, the pathway from adaptation to phase-leading response, and onto collective oscillations through signal relay, is firmly established (See "Methods"). Below we illustrate, with the help of three examples of increasing complexity, how this line of reasoning could link up different aspects of system behaviour to arrive at a renewed understanding. Implications of our model study to experimental work are given in the "Discussion" section.

**A weakly nonlinear model with adaptation**. We consider first a noisy two-component circuit which is a variant of the model for sensory adaptation in *E. coli*[41] (Fig. 3a, see also "Methods"). For weak noise, the intracellular signal relay in the quiescent state is essentially linear with the receptor response function given by

$$\tilde{R}_a(\omega) = \alpha_2 \left[ 1 + \frac{\epsilon}{\epsilon^2 + (\tau_y \omega)^2} + i\tau_a \omega^* \frac{(\omega^*/\omega) - (\omega/\omega^*)}{1 + (\epsilon/(\tau_y \omega))^2} \right]^{-1},$$

(10)

where

$$\omega^* = (\tau_a \tau_y)^{-1/2} (1 - \epsilon^2 \tau_a / \tau_y)^{1/2}.$$

(11)

Here $\tau_a$ and $\tau_y$ are the timescales for the activity ($a$) and negative feedback ($y$) dynamics, respectively. In Figs. 3b, c, we show the phase shift $\phi_a(\omega)$ and the real and imaginary part of $\tilde{R}_a(\omega)$ against the frequency $\omega$, plotted on semi-log scale. As predicted, $\phi_a(\omega)$ undergoes a sign change at $\omega^*$. Correspondingly, the imaginary component of the response $\tilde{R}_a''$ becomes negative in the phase-leading regime, violating the FDT. The peak of $|\tilde{R}_a(\omega)|$ is located close to $\omega^*$, with a relative width $\Delta\omega/\omega^* \simeq Q^{-1}$ where $Q = \tau_a \omega^* \simeq (\tau_a/\tau_y)^{1/2}$.

Allowing the chemoreceptor activity $a$ to affect the signal as in Eq. (1) with $F(s) = -Ks$, we observe an oscillatory phase upon increase in cell density in numerical simulations (Fig. 3d). Figure 3e shows the oscillation amplitude (upper panel) and frequency (lower panel) against the coupling strength $\overline{N}$ around the onset of oscillations. The threshold coupling strength $\overline{N}_o = N_o \alpha_1 \alpha_2$ and the onset frequency $\omega_o$ both agree well with the values predicted by Eqs. (4)–(5) (see arrows in Fig. 3e). The transition is well described by a supercritical Hopf bifurcation. At finite oscillation amplitudes, there is a downward shift of the oscillation frequency which can be quantitatively

calculated in the present case by introducing a renormalised response function $\tilde{R}_a^+(\omega)$ (see Supplementary Note 2), whose phase is shown in Fig. 3f. The oscillation frequency is determined by the crossing of the two curves $\phi_s(\omega)$ and $\phi_a^+(\omega)$, with the formal independent of the oscillation amplitude $A$. As the oscillation amplitude grows further, higher order harmonics generated by the nonlinear term become more prominent. The model system eventually exits from the limit cycle through an infinite-period bifurcation and arrives at a new quiescent state. The upper bifurcation point $\overline{N}_b$ is inversely proportional to the adaptation error $\epsilon$ (see Supplementary Fig. 1).

The signal phase shift $\phi_s(\omega)$ is given by Eq. (6). When the signal relaxation time $\tau_s$ is much shorter than the cell adaptation time $\tau^* \equiv 2\pi/\omega^*$, $\phi_s(\omega)$ stays close to zero so that the selected period is essentially given by $\tau^*$. In this case, $|\tilde{R}_a(\omega)|$ is near its peak and hence the cell density required by Eq. (5) is the lowest. As signal clearance slows down, the crossing point shifts to lower frequencies. Given a finite adaptation error $\epsilon > 0$, there is a generic maximum signal relaxation time $\tau_s^* \sim \epsilon^{-1}$ beyond which the phase matching cannot be achieved (see Supplementary Note 3 and also Supplementary Fig. 3).

**Excitable dynamics**. DQS in *Dictyostelium* and other eukaryotic cells takes the form of pulsed release of signalling molecules[2,7,51]. The highly nonlinear two-component FitzHugh-Nagumo (FHN) model is often employed for such excitable phenomena[3,52,53]. Similar to the sensory adaptation model discussed above, each FHN circuit has a memory node $y$ that keeps its activity $a$ low (the resting state) under a slow-varying signal $s(t)$ (Fig. 4a, see also "Methods"). On the other hand, a sufficiently strong noise fluctuation or a sudden shift of $s$ sends the circuit through a large excursion in phase space (known as a firing event) when $y$ is slow (i.e., $\tau_y \gg \tau_a$). Our numerical investigations show that the noise-triggered firing does not disrupt the adaptive nature of the circuit under the negative feedback from $y$. The noise-averaged response of a single FHN circuit exhibits the same characteristics as the sensory adaptation model, including adaptation to a stepwise stimulus after a transient response (Fig. 4b, upper panel), as well as the phase-leading behaviour and diminishing response amplitude on the low frequency side (Fig. 4b, lower panel).

Fig. 4c shows time traces of individual cell activities (blue and green curves) as well as that of the signal $s$ (red curve) from simulations of weakly coupled FHN circuits at three different values of the coupling strength $\overline{N}$ (see Methods). At $\overline{N} = 0.5$, the two selected cells fire asynchronously while $s$ remains constant. At $\overline{N} = 0.9$, collective behaviour as seen in the oscillation of $s$ starts to emerge, although individual circuits continue to fire sporadically. Upon further increase of $\overline{N}$, synchronised firing is

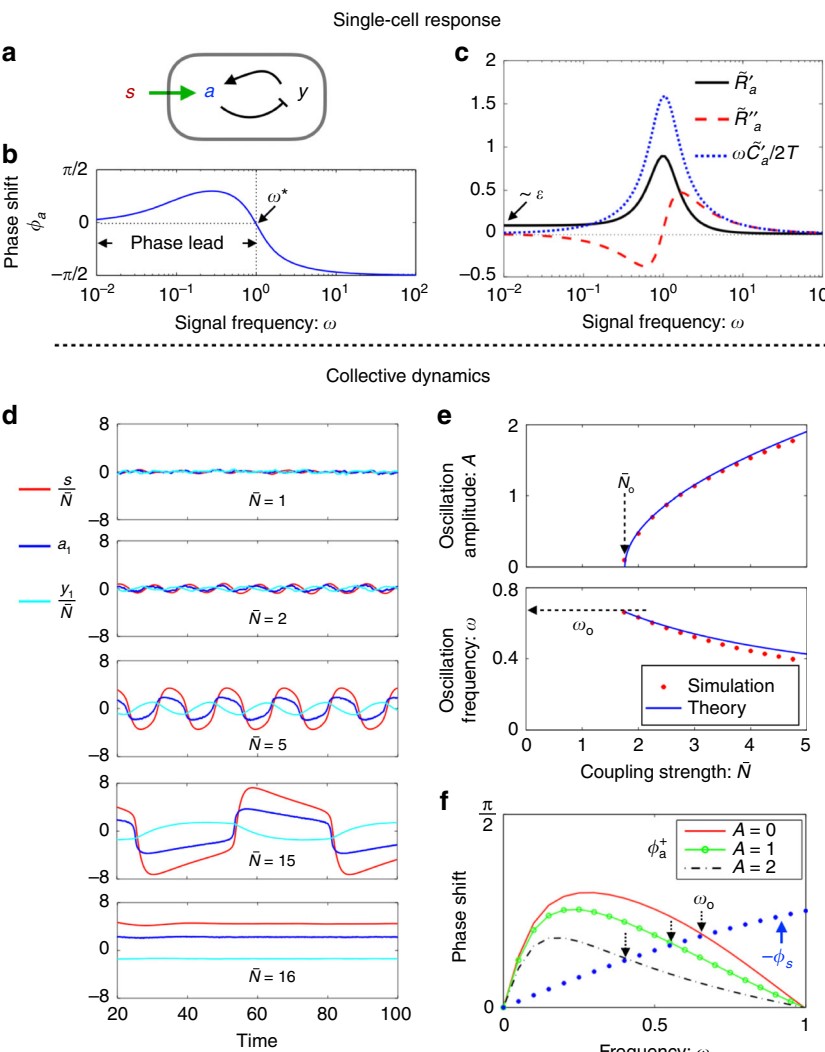

**Fig. 3 A weakly nonlinear model with adaptation. a–c** Single-cell response. **a** A noisy two-component model with negative feedback. **b** Frequency-resolved phase shift $\phi_a = -\arg(\tilde{R}_a)$. A sign change takes place at $\omega = \omega^* \simeq (\tau_a \tau_y)^{-1/2}$, with $a$ leading $s$ on the low frequency side. **c** Real ($\tilde{R}'_a$) and imaginary ($\tilde{R}''_a$) components of the response spectrum. $\tilde{R}'_a$ is of order $\epsilon$ in the zero frequency limit, while $\tilde{R}''_a$ changes sign at $\omega = \omega^*$. Also shown is the correlation spectrum $\tilde{C}_a(\omega)$ multiplied by $\omega/(2T)$, where $T$ is the noise strength. The fluctuation-dissipation theorem $\tilde{R}''_a = \omega \tilde{C}_a(\omega)/(2T)$ for thermal equilibrium systems is satisfied on the high frequency side, but violated at low frequencies. **d–f** Simulations of coupled adaptive circuits. **d** Time traces of the signal (red) and of the activity (blue) and memory (cyan) from one of the participating cells at various values of the coupling strength $\bar{N} = \alpha_1 \alpha_2 N$. **e** The oscillation amplitude $A$ (of activity $a$) and frequency $\omega$ against $\bar{N}$. The amplitude $A$ grows as $(\bar{N} - \bar{N}_o)^{1/2}$ here, a signature of Hopf bifurcation. **f** Determination of oscillation frequency from the renormalised phase matching condition at finite oscillation amplitudes: $\phi_a^+(\omega, A) = -\phi_s^+(\omega, A)$. The linear model for $s$ yields $\phi_s^+(\omega, A) = -\phi_s(\omega)$. Parameters: $\tau_a = \tau_y = \gamma = K = c_3 = 1$, $\alpha_1 = \alpha_2 = 0.5$, and $\epsilon = 0.1$. The strength of noise terms is set at $T = 0.01$.

established. Despite the highly nonlinear nature of the FHN model, both the onset coupling strength $\bar{N}_o$ and the frequency $\omega_o$ are well predicted by Eqs. (4)–(5) using the respective response functions in the resting state (Fig. 4d).

**Yeast glycolytic oscillations.** We take the adapt-to-oscillate scenario one step further to examine the dynamics of ATP auto-catalysis in yeast. Concentration oscillations of NADH and glycolytic intermediates have been observed in yeast cell extracts as well as in starved yeast cell suspensions upon shutting down the respiratory pathway (see ref. [22] for a review). The phospho-fructokinase (PFK), an enzyme in the upper part of the glycolytic pathway, is tightly regulated by ATP, a key product of glycolysis. This robust negative feedback is commonly regarded as the driver of glycolytic oscillations, with a typical period of 30–40 s in intact cells but 2 min or longer in extracts. Cells at high density oscillate

synchronously due to redox signalling via the freely diffusing molecule acetaldehyde (ACE)[22,26]. As the cell density decreases, the synchronised behaviour breaks down. While many studies found continued oscillation of individual cells at their own frequencies[25,54], simultaneous disappearance of individual and collective oscillations as in other DQS systems has also been reported[23]. We show below that both types of behaviour could be accommodated in a model of glycolysis that couples intracellular redox state to ATP autocatalysis.

We first investigate the dynamic properties of single-cell glycolysis at fixed intracellular glucose and ACE concentrations. Collective oscillations in yeast cell suspensions, which require ACE transport across the cell membrane, will be discussed later. Our starting point is the du Preez et al. model[55] that includes around 20 metabolic reactions (Fig. 5a). By monitoring the temporal response of metabolites under perturbations of the intracellular ACE concentration, we obtained a phase diagram

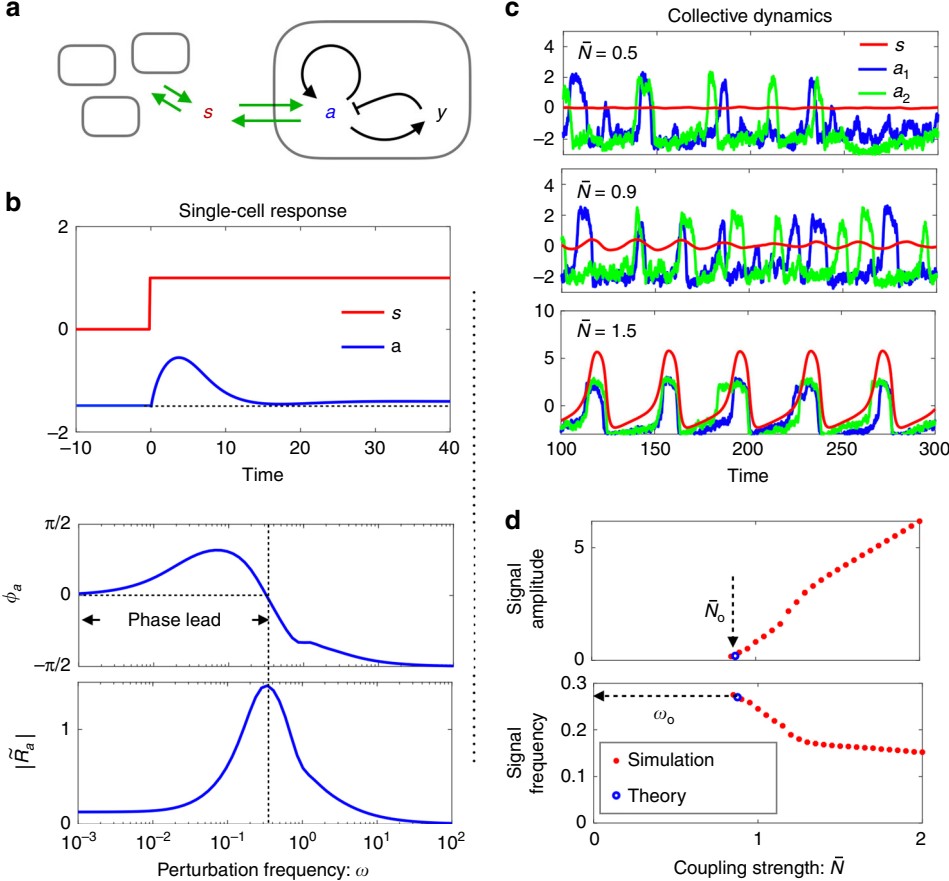

**Fig. 4 Simulations of the coupled excitable FitzHugn-Nagumo (FNH) model with noise. a** Model illustration. Note the self-activation of $a$ that gives rise to excitability (see Methods for details). **b** Noise-averaged response of $a$ in the resting state. Upper panel: the average response to a step signal. Lower panel: the response amplitude and phase shift at various signal frequencies. **c** Trajectories of the coupled FHN model at various values of the effective coupling strength $\overline{N}$. In addition to the signal $s$, activities of two out of a total of 1000 cells are plotted. **d** Signal oscillation amplitude and frequency against effective cell density. Red stars: simulation data; Blue circles: predictions of Eqs. (4)-(5) using numerically computed response spectra.

shown in Fig. 5b. The white region marks spontaneous oscillations in an isolated cell[56]. The glucose concentration, which controls glycolytic flux, needs to be sufficiently high for oscillations to take place. ACE also has a role in the dynamics: either very low or very high concentrations arrest the oscillations.

The non-oscillatory part of the phase diagram can be further divided into three sub-regimes according to the adaptive properties of the metabolic network against an ACE signal (coloured bands in Fig. 5b). Figure 5c gives, under one representative condition in each band, the concentration variation of four metabolites upon a sudden shift in the intracellular ACE concentration $ACE_{in}$. In all cases, the intracellular redox agent NAD follows closely ACE concentration change and hence acts as an instantaneous transducer of the signal. ATP adapts in both the orange and blue band, while PYR, the substrate to produce ACE, adapts only in the blue band. TRIO, the metabolite immediately upstream of the reaction GAPDH that uses NAD as cofactor, does not adapt. Overall, the adaptation error increases progressively as one moves away from the oscillatory region. A sign change in the response of ATP (and also of TRIO) takes place across the dashed line at $ACE_{in,0} \simeq 0.2$ mM.

Figure 5d shows phase shifts of ATP, NAD and five other metabolites along the glycolytic pathway against a periodic ACE signal at various frequencies. At the point marked by star in the orange band, ATP, BPG and PEP are phase-leading (after a $\pi\pi$ shift) below the frequency $\omega^* \simeq 20$ min$^{-1}$ (upper panel). The list is expanded to all six metabolites (except NAD which is

synchronised with the signal) when the environment shifts to the point marked by diamond in the blue band (lower panel). In particular, PYR (blue line) have a large phase lead around $\omega^*$, which matches its adaptive behaviour under Fig. 5c (Supplementary Note 4).

To further disentangle dynamical properties of the network, we constructed a reduced model in Fig. 5e by taking into account stoichiometry and known regulatory interactions along the glycolytic pathway[24], and by making use of the timescale separation in the turnover of metabolites as suggested by their response spectra (Fig. 5d and Supplementary Note 5). Since ATP and PYR now appear as co-products of the condensed reaction PYK in the reduced model, the latter can be viewed as a reporter of ATP homeostasis implemented by the negative feedback loop (cyan line in Fig. 5e). Figure 5f shows the phase diagram of the reduced model against the intracellular $ACE_{in,0}$. Similar to the full model at high glucose concentrations, the circuit enters an oscillatory state at intermediate values of $ACE_{in,0}$, and shows adaptive response on the two wings. The extended adaptive regime on the high $ACE_{in,0}$ side differs from the behaviour seen in Fig. 5b, but is reproduced by a mutant of the full model where the glyoxylate shunt (GLYO) is turned off (Supplementary Figs. 10–11).

We now examine a model of yeast cell suspensions where individual cells metabolise according to the reduced model and communicate their redox state through ACE (Supplementary Note 6). ACE is synthesised internally and degraded at rates $k_{in}$ and $k_{ex}$ within and outside the cell, respectively. The rate of its

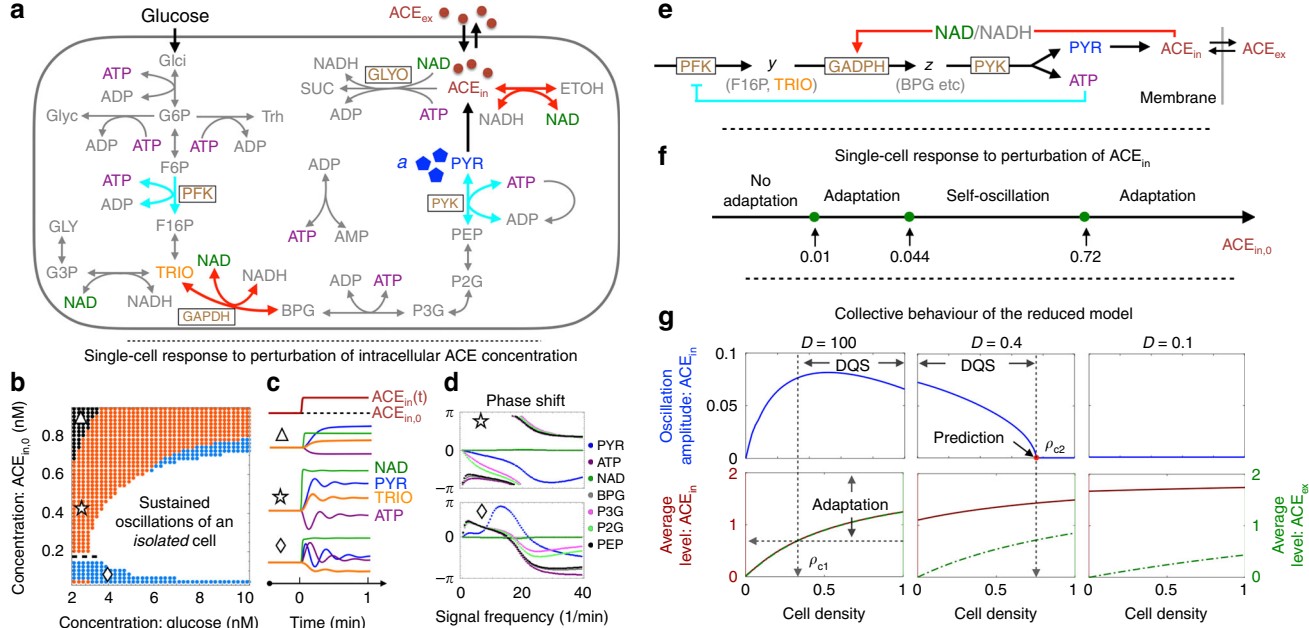

**Fig. 5 Yeast glycolytic oscillations. a** The reaction network of glycolysis in a yeast cell (see Supplementary Figure 3 for full names of the abbreviations). **b** Single-cell phase diagram spanned by the intracellular glucose and acetaldehyde (ACE) concentrations. Adaptation becomes less accurate further away from the oscillatory region (white), as indicated by coloured bands. The response of ATP changes sign around $ACE_{in,0} \simeq 0.2$ as indicated by the dashed line. **c** Representative time traces of metabolites to an upshift in intracellular ACE concentration ($ACE_{in}$) for selected conditions in each band (triangle, star, and diamond in (**b**)). **d** Frequency-resolved phase shifts of selected metabolites to weak sinusoidal perturbations of $ACE_{in}$. ATP, BPG and PEP are phase-leading in both blue and orange bands of the phase diagram, while PYR does so only in the blue band. **e** A reduced model for glycolytic oscillations where the intracellular NAD/NADH ratio and pyruvate (PYR) act as the receiver and sender of the signal (ACE), respectively. Adaptive response of PYR to ACE is coupled to the homeostasis of ATP through the reaction PYK. **f** Phase diagram of the reduced model against intracellular ACE concentration. **g** Oscillation amplitudes (upper panel) and time-averaged intracellular and extracellular ACE concentrations (lower panel) against cell density in a population where individual cells metabolise according to the reduced model. Data for three selected values of ACE membrane permeability $D$ are shown.

cross-membrane transport is set by the membrane permeability $D$. When $D$ is large, the intracellular and extracellular ACE levels are essentially the same (left panel, Fig. 5g). On the low density side, the homogeneous cell population enters the oscillatory phase through synchronisation of oscillatory cells. This behaviour continues beyond the point $ACE_{in} = 0.72$ (arbitrary unit) when an isolated cell switches from oscillation to adaptation (Fig. 5f). In other words, the system crosses over smoothly from oscillator synchronisation to adaptation-driven oscillations, or DQS. As $D$ decreases and becomes comparable to ACE degradation rates (set at $k_{in} = 0.5$ and $k_{ex} = 0.3$), the intracellular ACE concentration grows and eventually exceeds the upper self-oscillatory threshold $ACE_{in} = 0.72$ even at the low cell density limit. Our simulations indicate that DQS persists at $D = 0.4$ but disappears at $D = 0.1$ (middle and right panels, Fig. 5g). Interestingly, DQS at $D = 0.4$ disappears at an upper threshold density $\rho_{c2} = 0.73$. This inverse DQS, i.e., oscillation quenching at high cell density, can be quantitatively explained by Eqs. (4)–(5) using the numerically determined response functions that depend on the cell density (Supplementary Note 6 and Supplementary Fig. 15).

## Discussion

In this work, we investigated a general scenario for emerging oscillations in a group of cells that communicate via a shared signal. It covers a broad class of pulsation behaviour in cell populations, collectively known as DQS. Using the single-cell response to external stimulation, we formulated a quantitative requirement for the onset of collective oscillations that must be satisfied by active cells as well as models of them. A proof is presented to link this requirement to the adaptive release of signalling molecules by individual cells. Our work thus consolidates observations made

in the literature and formalises adaptation as a unifying theme behind DQS.

The above mathematical results connect well to the recent surge of interest in active systems, where collective phenomena emerge due to energy-driven processes on the microscopic scale[57,58]. The study of such non-equilibrium processes opens a new avenue to explore mechanisms of spontaneous motion on large scales. We presented a general formula for the energy outflow of a living cell through a designated mechanical or chemical channel under periodic stimulation. This energy flux is positive over a range of frequencies when the cell responds to the stimulus adaptively. Since adaptation is a measurable property of a cell, the thermodynamic relation is applicable without making specific assumptions about intracellular biochemical and regulatory processes, while most models do. When cells are placed together in a fixed volume, a quorum is required to activate the energy flow via self and mutual stimulation.

We reported three case studies to illustrate how these general yet quantitative relations could be applied to analyse the onset of collective oscillations in specific cell populations. Our first example is a coarse-grained model where signal reception and release are integrated into the same activity node (e.g., a membrane protein or a molecular motor). Due to the weak non-linearity of the intracellular circuit, many analytical results were obtained. The intracellular adaptive circuit has two timescales: the activity relaxation time $\tau_a$ and the negative feedback time $\tau_y$. Their ratio $Q^2 = \tau_a/\tau_y$, similar to the quality factor in resonators, determines the shape of the adaptive response (Fig. 2). At small adaptation error $\epsilon \ll 1$, the imaginary part of the response function $\tilde{R}_a(\omega)$ changes sign at the characteristic frequency $\omega^* \simeq (\tau_a \tau_y)^{-1/2}$. This is also approximately the frequency where

$|\tilde{R}_a(\omega)|$ reaches its maximum. When cells are coupled through the signal with a relaxation time $\tau_s$, the onset oscillation frequency $\omega_o$ increases with decreasing $\tau_s$, reaching its maximal value $\omega^*$ when $\tau_s \ll 1/\omega^*$.

Much of these results carry over to our second example, a population of coupled excitable circuits described by the FHN model. Despite its highly nonlinear nature, the FHN model in the resting state shows adaptive response under weak stimulation. Our numerical simulations of the coupled system at weak noise confirm the onset oscillation frequency and the critical cell density predicted by Eqs. (4)–(5).

The above theoretical predictions compare favourably with available experimental data. The first is mechanical stimulation of hair cells carried out by Martin et al.[19], where the cellular response was extracted using a flexible glass fibre. Deformation of the glass fibre, which is the signal here, has a relaxation timescale (~0.5 ms) much shorter than the adaptation time of the hair bundle (~0.1 s). Spontaneous oscillations of the combined system were observed at 8 Hz, the predicted frequency where the imaginary part of the hair bundle response function $\tilde{R}_a''(\omega)$ undergoes the expected sign change. The second is a recent microfluidic single-cell measurement of *Dictyostelium* reported by Sgro et al.[3], where the change of cytosolic cAMP level (activity $a$) in response to extracellular cAMP variation (signal $s$) was presented. From the measured response $a(t)$ to a step increase of the signal in their work (reproduced in Fig. 6a, upper panel), we computationally deduced the response function $R_a(t) = da/dt$ in the time domain (Fig. 6a, lower panel) and then the response spectrum $\tilde{R}_a$ via Fourier transform. The resulting phase shift $\phi_a$ changes sign around $\omega^* = 1 \text{ min}^{-1}$ (Fig. 6b). According to our theory, the onset oscillation period at high flow rate should be around 6.28 min, which is indeed what was observed in experiments[2–4].

DQS in *Dictyostelium* is a time-dependent phenomenon coupled to cell migration and development[1,5]. In the experiments reported in refs. [2,3], synchronised firing of cells starts five hours after nutrient deprivation. The period of firing shortens from 15–30 min at the onset to 8 min and thereafter 6 min as cells begin to aggregate. Therefore the onset of collective oscillations may not be triggered by a critical cell density as such but the cell density does affect the period of oscillations. Previously, a property known as fold-change detection (FCD) was invoked and verified to explain cell-cell signalling even when cells are far apart[4]. In FCD, the intracellular signal relay circuit is activated by a relative change $\Delta s/s$ of the signal $s$. Consequently, the detection

sensitivity of the activity response function $\alpha_2 \sim 1/s$. In a population of communicating cells, the signal strength $s$ is proportional to the cell density $N$. Hence, FCD renders the signal relay efficiency $\overline{N}$ independent of the cell density $N$. To explain the accelerated pulsing at increasing cell densities, other aspects of the system need to be considered, e.g., cAMP clearance by phosphodiesterase secreted by cells[61]. Building these details into the FHN model, Sgro et al.[3] showed that the coupled equations are able to qualitatively reproduce the observed behaviour. The data analysis procedure illustrated by Fig. 6 offers a direct way to link pulsation from 8 min to about 6 min with a faster signal clearance effected by a higher concentration of phosphodiesterase in the surrounding medium. The long firing interval at early stage of the development could be attributed to physiological differences in the intracellular molecular network, e.g., a much longer negative feedback time $\tau_y$ that awaits experimental verification[59,60]. With this type of data, a similar procedure could be applied to analyse the segmentation clock in the presomitic mesoderm[7].

Our third example, the glycolytic oscillation in yeast cell suspensions, is also an open problem. Simulation studies of a detailed model of yeast glycolysis[55] yielded a relatively simple phase diagram shown in Fig. 5b, with the intracellular glucose and ACE concentrations as control parameters. As reported previously[55], cells in the white region oscillate spontaneously in a constant environment, driven by an instability associated with the negative feedback in ATP autocatalysis. In the neighbourhood of this region, we found that the ATP concentration adapts to the intracellular environment, in particular to a sudden shift in ACE concentration that affects directly the intracellular NAD/NADH ratio. The adaptation error increases as one moves away from the oscillatory region. These dynamical features are captured by a reduced model of ATP autocatalysis we proposed to approximate the low-dimensional attractor of the full model at high glucose concentrations. We then considered a homogeneous population of cells that carry out fermentation according to the reduced model, using membrane permeability $D$ to tune intracellular ACE concentration at a given cell density. When $D$ is much greater than the ACE turnover/degradation rates, the intracellular and extracellular ACE levels are equilibrated. In such a situation, collective oscillations on the low cell density side first emerge through synchronisation of individual cells that enter the self-oscillatory state. Further increase of the cell density elevates both intracellular and extracellular ACE levels, eventually brings individual cells out of the self-oscillatory state. However, the

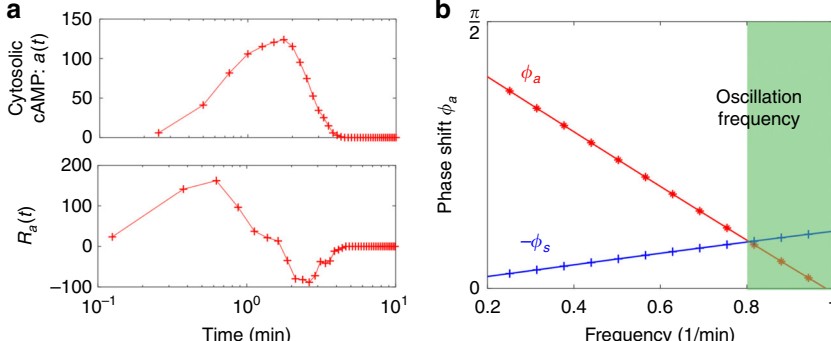

**Fig. 6 Intracellular activity response function constructed from single-cell measurements on *Dictyostelium*. a** Upper panel: The average cytosolic cAMP level (the activity $a$, arbitrary unit) in response to a step increase of 1 nM extracellular cAMP at $t = 0$ (reproduced from Fig. 2a in ref. [3]). Lower panel: The response function $R_a(t)$ estimated from the derivative of the response data in the upper panel. **b** The corresponding phase shift $\phi_a$ in the low frequency regime (red curve), obtained from the Fourier transform of $R_a(t)$. Onset oscillation frequencies in experiments reside in the green region (see Fig. 2b in ref. [2]). Also shown is the phase shift $-\phi_s$ from Eq. (6) at $\tau_s = 0.35$ min (blue curve), the extracellular signal clearance time to yield an onset oscillation period of about 8 min. Faster signal clearance shortens the oscillation period towards the theoretical lower bound at around 6.28 min.

population continues to oscillate following the DQS scenario. Slower cross-membrane diffusion drives up intracellular ACE level and, at some point, eliminates self-oscillation. Nevertheless, the population may still oscillate via DQS at an intermediate range of cell densities. Beyond an upper critical cell density, the diminishing adaptive response of glycolytic flux to the signal eventually arrests collective oscillations. This new phenomenon, which we name inverse DQS, is quantitatively predicted by our theory. We note that the strong cell-to-cell variability observed in single-cell experiments[56] could significant alter the behaviour shown in Fig. 5g on the low cell density side, an issue we leave to future work.

These model studies helped to refine and resolve various quantitative issues in the induction of collective oscillations in well-studied systems, and at the same time inspire novel applications built around adaptation-driven signal relay. One promising direction to follow is the development of artificial oscillatory systems with techniques from synthetic biology[62–65]. In analogy with the hair cell/glass fibre setup, one may think of tricking a quorum-sensing cell to oscillate by confining it to a volume small enough to enable positive signal relay.

In statistical physics, the response function formalism is widely used to analyse system level response to environmental perturbations, but its application to collective behaviour in biological systems is still limited. Our examples show that cell models with different levels of biological detail, out of either necessity or convenience, could yield qualitatively or even quantitatively similar response curves with respect to, say the production of a particular chemical used in cell-to-cell communication, which is reassuring. As these curves are increasingly accessible from experiments, their direct use for analysis and hypothesis building is highly desirable. With respect to the link between adaptation and collective oscillations, our formulation unifies and generalises previous studies in at least three specific settings. The first is an abstract 3-variable model that connects fold-change detection of individual cells to the robustness of collective oscillations over a broad range of cell densities[4]. In the second case, adaptation was proposed to play an important role in the collective oscillation of neuronal networks[49]. Lastly, an Ising-type model of chemoreceptor arrays in *E. coli*[50] predicts that increasing the coupling strength between adaptive receptors drives the system to collective oscillations, although in reality the chemoreceptor array manages to operate below the oscillatory regime. Despite the risk of running into an oscillatory instability, the coupling enhances sensitivity of the array to ligand binding. Along this sensitivity-stability tradeoff, one may speculate that some of the reported collective oscillations under laboratory conditions could actually arise from over perfection of adaptive/homeostatic response in the natural environment, a hypothesis that invites further experimental testing.

## Methods

**An adaptive model with cubic nonlinearity.** The data presented in Fig. 3 were obtained from numerical integration of the coupled equations[41,44]: $\tau_a \dot{a} = -a - c_3 a^3 + y + \alpha_2 s + \eta_a$, and $\tau_y \dot{y} = -a - \epsilon y + \eta_y$. Here $y$ is a memory node that implements negative feedback control on $a$, $\epsilon$ sets the adaptation error, and $\tau_a$ and $\tau_y$ are the intrinsic timescales for the dynamics of $a$ and $y$, respectively. $\eta_a$ and $\eta_y$ are gaussian white noise with zero mean and correlations: $\langle \eta_a(t)\eta_a(\tau)\rangle = 2T\tau_a \delta(t-\tau)$ and $\langle \eta_y(t)\eta_y(\tau)\rangle = 2T\tau_y \delta(t-\tau)$, where $\delta(t)$ is the Dirac delta function. The cubic nonlinearity ($c_3 a^3$) is needed to limit cellular activity to a finite strength. For simplicity, we choose $\alpha_2 = 1$ so that the response function defined by $\tilde{R}_a(\omega) = \langle \tilde{a}(\omega)\rangle / \tilde{s}(\omega)$ can be compared with its equilibrium counterpart that satisfies the FDT $\tilde{R}_a'' = \omega \tilde{C}_a(\omega)/(2T)$, with $\tilde{R}_a''$ denoting the imaginary component of $\tilde{R}_a$. Data in Fig. 3 were obtained by coupling cells via Eq. (1) with $F(s) = -Ks$ and $\xi = 0$.

**Existence of oscillatory state under an adaptive response.** We have shown in the Main Text that adaptive intracellular observables exhibit a phase-leading response in a certain frequency interval. For a given adaptive observable $a$, the phase lead $\phi_a(\omega)$ spans a continuous range from 0 to a maximum value $\phi_a^{\max}$ ($< \pi$). Meanwhile, the phase delay $\phi_s = -\tan^{-1}(\omega\tau_s)$ varies continuously from 0 to $-\pi/2$ (Eq. (6)). Since $\tau_s$ controls how fast $\phi_s(\omega)$ decreases from 0 to $-\pi/2$ as $\omega$ increases, intersection of $-\phi_s(\omega)$ with $\phi_a(\omega)$ can always be found by tuning $\tau_s$. In particular, when $\tau_s \to 0$, a solution is found at the high frequency end of the active frequency interval where $\phi_a(\omega) = 0$. From this discussion, we see that the onset frequency $\omega_o$ of oscillations is mostly determined by the intracellular dynamics, i.e., $\phi_a(\omega)$, but the medium can have a weak effect on $\omega_o$ when its relaxation time is comparable to that of the intracellular dynamics.

**Coupled FHN model.** A single FHN circuit takes the form, $\tau_a \dot{a}_j = a_j - a_j^3/3 - y_j + \alpha_2 s + \eta_{a_j}$, $\tau_y \dot{y}_j = a_j - \epsilon y_j + a_0 + \eta_{y_j}$. The positive sign of the first term in the equation for $a_j$ gives rise to excitability. In the absence of the stimulus $s$, each cell assumes the resting state with a mean activity $a_{rs} = \langle a_j(t)\rangle$. For small values of $\epsilon$, the resting state activity $a_{rs} \simeq -a_0$ is nearly constant under a slow-varying $s(t)$. FHN circuits are coupled together through a signal field whose dynamics is described by, $\tau_s \dot{s} = -s + \alpha_1 \sum_j^N (a_j - a_{rs})$. The parameters used in generating Fig. 4 are: $\alpha_2 = 1$, $N = 1000$, $\epsilon = 0.1$, $T = 0.1$, $\tau_a = 1$, $\tau_y = 5$, $a_0 = 1.5$, and $\tau_s = 1$. $\alpha_1 = \overline{N}/(N\alpha_2)$ is determined by the control parameter $\overline{N}$.

**Reporting summary.** Further information on research design is available in the Nature Research Reporting Summary linked to this article.

## Data availability
The data that support the findings of this study are available from S.-W. Wang on request. They can also be generated from the provided code.

## Code availability
The code that support the findings of this study are available at https://github.com/ascendancy09/Collective-oscillations.

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

## Acknowledgements
The authors thank Allon Klein and Kyogo Kawaguchi for helpful discussions and suggestions on the manuscript. The work is supported in part by the NSFC under Grant Nos. U1430237, 11635002 and U1530401, and by the Research Grants Council of the Hong Kong Special Administrative Region (HKSAR) under Grant No. 12301514 and C2014-15G.

## Author contributions
S.-W.W. and L.-H.T. designed research, performed research, analysed data, and wrote the paper.

## Competing interests
The authors declare no competing interests.
