## [Peer Review File · Nature Communications]

Reviewers' comments:

Reviewer #1 (Remarks to the Author):

In the manuscript by Wang and Tang on "Emergence of collective oscillations in adaptive cells", the authors studied the general requirements for a population of cells to develop synchronized oscillatory behavior by responding to a common pool of signaling molecules that are produced and secreted by the cells themselves. The authors call this phenomenon "dynamical quorum sensing" (DQS). This idea is then used to study specific biological systems such as the Yeast glycolysis pathway. There are three explicitly stated "main results" in the manuscript, which I comment on one by one in the following:

1. The first main result is Eq. 3, which stated the condition for the onset of collective oscillations in the system where the cells (a) and the signal (s) are coupled to each other reciprocally through different response functions R_a and R_s . From Eq. 3, the authors draw their first conclusion that the (active) cells need to respond to the signal with a phase advance in order to achieve self-sustained oscillation. Although Eq. 3 itself is just the result of linear (stability) analysis for a simple coupled linear ODE's (Eq.1 and Eq. 2) in frequency space, the consequence, in particular, the phase-leading requirement for the active cells is nonetheless an interesting general observation.

2. The second result of the paper is that oscillatory signaling dissipates energy. As it is presented in the manuscript, I am not sure how important a result this is. The fact that biochemical oscillations cost energy is very well established, so it is clear what is exactly new here.

3. The third main result, which is also stated as "the most important result in our paper", is about the connection between phase leading behavior, which is necessary for oscillation, and adaptive response of the system. Based on the general Kramers-Kronig relation (Eq. 4) that applies for the response function that is analytic in the upper plane due to causality, the authors conclude that the cells that adapt accurately are susceptible to oscillations. Although this result is more general, the fact that adaptive systems can generate oscillatory behavior has been studied before including the case in *E. coli* chemotaxis which is used as an example in this manuscript. For example, in a paper by Mello et al (Biophysical Journal, vol 87, 2004), it was shown that in an Ising-type model of *E. coli* chemoreceptor array, the system goes from being adaptive to oscillatory as the receptor-receptor coupling strength, which is directly related to the gain of the system, increases beyond a critical value. Although the detailed mechanisms may be different here from those studied before, their connections as well differences should be addressed carefully here.

Besides the above comments with respect to the "main results" of the paper, I also have a couple of additional questions:

4. What is the general structure of the response function R_a in real time space? Is there a typical time scale(s) associated with it? In Fig. 2A, $\langle a \rangle$ was plotted, is it just $\langle a(t) \rangle$ as defined in Eq. 2?
5. Related to the above question, a_2 seems to be an important parameter in the response function as it appeared in the expression in the effective coupling strength N^- . However, a_2 was not defined in the paper mathematically so one can see how R_a depends on a_2 .

6. What is the physical meaning of the coupling strength N^- ? what sets its critical value for the onset of simulation? In the model, why does oscillation disappear when N^- becomes much bigger(=16) as shown in Fig. 4A?

7. In the simulation study of the Yeast glycolytic oscillations, if ACE is the signal, and if we believe the system follows the theory presented in this work, shouldn't we expect ACE to also oscillate? Even in the orange regime in Fig. 4B, where the oscillations of ATP occur in the model, they seem to be only damped oscillation. Why? Does the theory also apply to damped oscillation? Finally, does the theory provide any testable predictions to distinguish the two possible scenarios –

synchronization or adaptive response?

Overall, I believe this study presented some useful insights for understanding possible collective oscillatory behaviors in adaptive systems based on simple linear analysis in frequency space. The generality of the results is the most appealing point of the paper. However, some of the main results need to be flushed out better with respect to previous work and existing knowledge to highlight what is really new here. Technically, I think more systematic derivation of key results should be presented – this is a theory paper after all. Finally, some testable predictions need to be made to make it appeal to biologists.

Reviewer #2 (Remarks to the Author):

This paper first presents a high-level model and identifies two generic constraints for the onset of collective oscillation in a group of cells. Then, using non-equilibrium thermodynamics, the paper shows one of the constraints on group oscillations dissipates energy. The key link the authors get from this model is that the ubiquitous single cell adaptive behavior, in a relatively wide parameter space, satisfies the two constraints, and therefore can serve as a source for the emergence of collective oscillations. By providing a model analysis on yeast glycolysis network, the authors predict a new regime where a molecule in the network exhibits adaptive behavior, and they propose that apart from resulting from synchronized single cell oscillations, the collective oscillation could also be a result from adaptive response. Because the model is formulated as a generic response function, it can be adopted to describe both physical and chemical collective oscillations with different intracellular molecular networks. To show generality of the model, in the supplement of the paper, the authors analyze three different examples with the model framework and suggest that the adaptation/excitable response are what triggers the onset of group oscillation.

Major issues with regards to interest to a broad community and strengthening the conclusions:

- The authors need to work on demonstrating the novelty and the wide impact they claim in the paper. The only novel finding with the help of this model framework is that there might be a new adaptive regime that contributes to the collective yeast glycolysis oscillation. The authors should elaborate on what other novel conclusions can be reached and show how other models may fail to satisfy all of the requirements of published data that theirs does meet.
- Generality of the model is mostly demonstrated through the three examples buried in the supplement. The generality of the paper could be stressed more if the examples are elaborated in the main text. Indeed, the main text could be expanded from the text of the supplement to help clarify the novel points of the paper. The main text is so short that it is hard to support their claims of novelty.
- There exist many general models that have been used to shed light on potential origins of collective oscillations. This paper could be strengthened by discussing how one could potentially validate or invalidate the theoretical findings of the paper in experimental systems. That is, are there potential knockout experiments or experiments modulating cell density or physical/biochemical environment that would test the claims made here? The authors do specifically say that "In principle, even a single cell (i.e., $N = 1$) could oscillate under the identified mechanism when the coupling constants α_1 and α_2 are sufficiently large, e.g. the cell is confined to a small volume. It would be interesting to perform experiments on cells of different group size and analyze the data using the response function formalism developed here." However, what is interesting about this and what potential validation or breakdown of the model with these experiments would look like and provide that is useful beyond what is already known is not clarified.

Minor issues:

- In the paragraph about Fig S5B, the paper notes an opposite trend of frequency shift- coupling

relationship between model- predicted and actual data results, but just dropped the statement there. More discussion of this trend would be helpful in understanding why this potentially happens.

- In the paragraph discussing routes to collective oscillations in social amoeba, the paper presents two possible routes: adaptive response and single cell limit cycle oscillations. However, experimental work (cited in the paper as refs. 2 and 3) has demonstrated that in the collective natural environment, extracellular cAMP likely never reaches the limit cycle regime.

Reply to Referee A

We thank Referee A for handling our paper. His remarks have been highly valuable towards improving the clarity of our paper and strengthening connections with the past literature. Below we answer his comments one by one.

Comment 1: *The first main result is Eq. 3, which stated the condition for the onset of collective oscillations in the system where the cells (a) and the signal (s) are coupled to each other reciprocally through different response functions R_a and R_s . From Eq. 3, the authors draw their first conclusion that the (active) cells need to respond to the signal with a phase advance in order to achieve self-sustained oscillation. Although Eq. 3 itself is just the result of linear (stability) analysis for a simple coupled linear ODE's (Eq.1 and Eq. 2) in frequency space, the consequence, in particular, the phase-leading requirement for the active cells is nonetheless an interesting general observation.*

Answer 1: We thank the referee for appreciating our result as a very general and also interesting observation.

Comment 2: *The second result of the paper is that oscillatory signaling dissipates energy. As it is presented in the manuscript, I am not sure how important a result this is. The fact that biochemical oscillations cost energy is very well established, so it is unclear what is exactly new here.*

Answer 2: We apologise that the statement in the previous version is too vague. Indeed, it is very well established that biochemical oscillations cost energy. The novelty here is an interesting observation on the cell-to-signal energy flow when a cell is stimulated in the phase-leading frequency regime. In the revised manuscript, we moved Eq. (6), which states our result explicitly, from the Supplementary to the Main Text.

Comment 3: *The third main result, which is also stated as “the most important result in our paper”, is about the connection between phase leading behavior, which is necessary for oscillation, and adaptive response of the system. Based on the general Kramers-Kronig relation (Eq. 4) that applies for the response function that is analytic in the upper plane*

due to causality, the authors conclude that the cells that adapt accurately are susceptible to oscillations. Although this result is more general, the fact that adaptive systems can generate oscillatory behavior has been studied before including the case in *E. coli* chemotaxis which is used as an example in this manuscript. For example, in a paper by Mello *et al.* (*Biophysical Journal*, vol 87, 2004), it was shown that in an Ising-type model of *E. coli* chemoreceptor array, the system goes from being adaptive to oscillatory as the receptor-receptor coupling strength, which is directly related to the gain of the system, increases beyond a critical value. Although the detailed mechanisms may be different here from those studied before, their connections as well differences should be addressed carefully here.

Answer 3: We thank the referee for appreciating the generality of our result that links adaptation to phase-leading response, and also for reminding us of the earlier and nice work by Mello *et al.* In the revised manuscript, reference to previous work reporting the connection between adaptation and oscillatory behaviour is made in the paragraph immediately following Eq. (8). Furthermore, an expanded discussion is added at the end of the paper: “With respect to the link between adaptation and collective oscillations, our formulation unifies and generalises previous studies in at least three specific settings. The first is an abstract 3-variable model that connects fold-change detection of individual cells to the robustness of collective oscillations over a broad range of cell densities[4]. In the second case, adaptation was proposed to play an important role in the collective oscillation of neuronal networks[49]. Lastly, an Ising-type model of chemoreceptor arrays in *E. coli*[50] predicts that increasing the coupling strength between adaptive receptors drives the system to collective oscillations, although in reality the chemoreceptor array manages to operate below the oscillatory regime. Despite the risk of running into an oscillatory instability, the coupling enhances sensitivity of the array to ligand binding. Along this sensitivity-stability tradeoff, one may speculate that some of the reported collective oscillations under laboratory conditions could actually be perfection of adaptive/homeostatic response in the natural environment gone overboard, a hypothesis that invites further experimental testing.”

Comment 4: *What is the general structure of the response function R_a in real time space? Is there a typical time scale(s) associated with it? In Fig. 2A, $R_a(t)$ was plotted, is it just as defined in Eq. 2?*

Answer 4: We apologise for not being clear here. The response function $R_a(t)$ in real time is indeed defined by Eq. 2, while Fig. 2A gives two representative sketches of the noise-averaged excess activity Δa under a stepwise signal of strength Δs . According to Eq. 2, we have $\Delta a(t) = \Delta s \int_0^t d\tau R_a(\tau)$ or $R_a(t) = d[\Delta a(t)/\Delta s]/dt$. When a is adaptive, we have

$$\epsilon = \Delta a(\infty)/\Delta s = \int_0^\infty R_a(\tau) d\tau = \lim_{\omega \rightarrow 0} \tilde{R}'_a(\omega).$$

A small adaptation error ϵ requires $R_a(t)$ to have both positive and negative parts. The number of nodes of $R_a(t)$ depends on the system details. For the sensory adaptation model in the first part of the Methods section, the temporal structure of $R_a(t)$ is determined by two timescales, one associated with the activity dynamics (τ_a), and the other negative feedback (τ_y). $\tau_y > 4\tau_a$ corresponds to the monotonic decay of $a(t)$ in Fig. 2A (solid line), while $\tau_y < 4\tau_a$ corresponds to oscillatory decay. The Fourier transform $\tilde{R}(\omega)$ of $R_a(t)$ is now given explicitly in the Main Text (Eq. 9). Quite generally, $|\tilde{R}(\omega)|$ is peaked around the frequency ω^* (Eq. 10) where the imaginary part of $\tilde{R}(\omega)$ vanishes. The two timescale case is rather typical although adaptation with multiple timescales may in principle happen. Note also that $R_a(t)$ at short times approaches a finite constant under receptor dynamics, but vanishes when a is downstream of the signal receptor, as in Fig. 6.

Comment 5: *Related to the above question, α_2 seems to be an important parameter in the response function as it appeared in the expression in the effective coupling strength \bar{N} . However, α_2 was not defined in the paper mathematically so one cannot see how R_a depends on α_2 .*

Answer 5: We thank the referee for pointing out the missing explicit definition of the signal-sensing sensitivity α_2 that sets the overall amplitude of R_a . We now clarified this in the paragraph following Eq. (4) by saying that ‘Let $\alpha_2 \sim |\tilde{R}_a|$ be the sensitivity of the cell activity against s . We introduce a “signal relay efficiency” $\bar{N} \equiv N\alpha_1\alpha_2$ which sets the overall coupling strength of the population through the signal.’ An explicit formula for \tilde{R}_a is given for the weakly nonlinear model with adaptation [Eq. (9)], where α_2 is essentially the amplitude.

Comment 6: *What is the physical meaning of the coupling strength \bar{N} ? what sets its critical value for the onset of simulation? In the model, why does oscillation disappear when \bar{N} becomes much bigger(=16) as shown in Fig. 4A?*

Answer 6: We thank the referee for asking this important question. In essence, $\bar{N} = N\alpha_1\alpha_2$ describes the total efficiency of signal relay through the stimulated activity of N cells in a given volume. Given its prominent role in this paper, we now name it the “signal relay efficiency”. Following Eq. (4b), we mentioned in the revised main text that “Oscillations start at the critical coupling strength $\bar{N}_o = N_o\alpha_1\alpha_2$. Eq. (4b) simply states that, at the selected frequency ω_o , signal amplification through the collective action of the cells compensates signal loss from dissipative forces acting on s , e.g., friction for a mechanical signal or degradation/dilution for a chemical signal.” At large values of \bar{N} , our coupled system eventually develops a new fixed point at a finite activity. For the parameters chosen, transition into this state takes place around $\bar{N} = 16$ through an infinite-period bifurcation. In the revised manuscript, a sentence at the end of the paragraph after the one containing Eq. (10) is added: “The coupled system eventually exits from the limit cycle through an infinite-period bifurcation and arrives at a new resting state. The upper bifurcation point \bar{N}_b is inversely proportional to the adaptation error ϵ (See Supplementary Fig. S1).”

Comment 7: *In the simulation study of the Yeast glycolytic oscillations, if ACE is the signal, and if we believe the system follows the theory presented in this work, shouldn't we expect ACE to also oscillate? Even in the orange regime in Fig. 5B, where the oscillations of ATP occur in the model, they seem to be only damped oscillation. Why? Does the theory also apply to damped oscillation? Finally, does the theory provide any testable predictions to distinguish the two possible scenarios – synchronization or adaptive response?*

Answer 7: We are sorry for misleading the referee here. Fig. 5 only presents the model of glycolytic pathway within a yeast cell, and only results of single-cell response are reported. The damped oscillations seen in Fig. 5C are for the response of selected metabolites under a ramp in extracellular ACE concentration. In this system, the negative feedback timescale τ_y is significantly shorter than the activity relaxation time τ_a . By coupling our minimal model of glycolysis in individual cells, we indeed observe oscillations of ACE (Supplementary Fig. S11-S13), and also all the metabolite concentrations in the network. To avoid possible confusion, we inserted an embedded caption in Fig. 5 that says “Single-cell response to perturbations of ACE concentration” and changed the text within the white region in Fig. 8B to be “Sustained oscillations of an *isolated* cell”.

With regard to the ongoing debate on the two possible scenarios – synchronisation or

adaptive response, our simulations of a system of identical cells show collective oscillations over a broad range of cell densities. The situation on the low density side resembles more of the Kuramoto model while the one on the high density side may be attributed to adaptive response. Although a new finding, this alone does not allow one to settle the debate, which asks for the state of individual cells when the collective oscillation disappears. In the revised manuscript, we added the following sentence in the Discussion part: “In this respect, our study unifies the two competing scenarios regarding the origin of glycolytic oscillations, i.e., autonomous or through mutual stimulation. Cell-to-cell variability can change the size of each regime on the single-cell phase diagram. Furthermore, the characteristic frequency ω^* of each cell could have a significant spread in both the adaptive and oscillatory regimes[56]. Consequently, the range of cell densities where collective oscillations take place could be much reduced. More work is needed to see whether the two scenarios could be separated in a heterogeneous population of yeast cells by tuning the degradation/dilution rate of the extracellular ACE.”

Comment 8: *Overall, I believe this study presented some useful insights for understanding possible collective oscillatory behaviors in adaptive systems based on simple linear analysis in frequency space. The generality of the results is the most appealing point of the paper.*

Answer 8: We thank the referee for appreciating the generality of our work.

Comment 9: *However, some of the main results need to be flushed out better with respect to previous work and existing knowledge to highlight what is really new here.*

Answer 9: We agree with the referee about the deficiency in the previous version. With the extensive rewriting and addition of new material and discussions, we hope we have addressed this concern.

Comment 10: *Technically, I think more systematic derivation of key results should be presented – this is a theory paper after all.*

Answer 10: We completely agree that a clear presentation of key findings is very important. In the revised manuscript, additional key equations [say, Eq. (5-6) and Eq. (8-10)] are included to communicate our results more accurately. Combined with the figures and much expanded Discussion section, the reader should have a much easier time to work through the

manuscript and map out the main results without referring to the Supplementary material.

Comment 11: *Finally, some testable predictions need to be made to make it appeal to biologists.*

Answer 11: We thank the referee for pointing out this important issue. In this work, we have established that adaptive cells in a group transform themselves from quiescent to oscillating once certain physical conditions are satisfied. The physical conditions consist of an amplitude relation for the efficiency of intracellular signal relay, and a phase relation which is satisfied by adaptive signal relay and fast signal degradation in the medium. In the revised manuscript, we discussed these issues at length with reference to the two model systems: DQS in dicty and yeast glycolytic oscillations. Our main findings are listed below.

1) *DQS in dicty*. Our quantitative prediction on the oscillation frequency through phase matching agrees well with experiments performed by Thomas Gregor and collaborators (Fig. 6). That DQS requires a sufficiently high cAMP degradation rate in the medium is also borne out by previous experiments. Due to logarithmic sensing of the extracellular cAMP concentration, the adaptive intracellular cAMP relay circuit could support DQS over a broad range of cell densities, as demonstrated by previous experiments and model studies, and is also in agreement with our amplitude relation with $\alpha_2 \sim 1/s$. The actual spatiotemporal dynamics prior to and during aggregation is highly complex and takes several hours to complete. It is possible that, during this time, cells also change their gene expression and cAMP-related signalling, thus invalidating the assumption of a fixed response spectrum or a fixed set of model parameters. We leave these issues for future study.

2) *Yeast glycolytic oscillations*. Our computational studies show that the ATP auto-catalysis feedback circuit has an adaptive region (for extracellular-intracellular acetaldehyde relay) on the phase diagram spanned by extracellular glucose and acetaldehyde concentrations. The size of this region can be significantly expanded by turning off the glyoxylate shunt in order to strengthen the coupling between ACE signal and ATP homeostasis. For homogeneous cell populations, we predict glycolytic oscillations over a broader range of cell densities due to the adaptive response. Heterogeneities in the cell population and additional ACE clearance mechanisms in the medium can also be considered within our general formalism. Here, more computational work is needed to come up with specific predictions.

3) *Artificial oscillatory units*. As a direct consequence of the amplitude relation, even a

single adaptive cell could be induced to oscillate when confined in a sufficiently small volume. This could be considered as a prediction coming out of present work, although conceptually it is quite similar to the hair cell/glass fibre system studied previously in Hudspeth's lab.

The above points are incorporated in the revised version, particularly in the Discussion section.

Reply to Referee B

We thank Referee B for handling our paper. His remarks have been highly valuable towards improving the quality of our paper, making it more accessible to a broad audience. Below, we answer his comments one by one.

Comment 1: *This paper first presents a high-level model and identifies two generic constraints for the onset of collective oscillation in a group of cells. Then, using non-equilibrium thermodynamics, the paper shows one of the constraints on group oscillations dissipates energy. The key link the authors get from this model is that the ubiquitous single cell adaptive behavior, in a relatively wide parameter space, satisfies the two constraints, and therefore can serve as a source for the emergence of collective oscillations. By providing a model analysis on yeast glycolysis network, the authors predict a new regime where a molecule in the network exhibits adaptive behavior, and they propose that apart from resulting from synchronized single cell oscillations, the collective oscillation could also be a result from adaptive response. Because the model is formulated as a generic response function, it can be adopted to describe both physical and chemical collective oscillations with different intracellular molecular networks. To show generality of the model, in the supplement of the paper, the authors analyze three different examples with the model framework and suggest that the adaptation/excitability response are what triggers the onset of group oscillation.*

Answer 1: We thank the referee for a careful and critical reading of our manuscript, and for his positive assessment of our contribution.

Below, we answer “major issues with regards to interest to a broad community and strengthening the conclusions” raised by referee B.

Comment 2: *The authors need to work on demonstrating the novelty and the wide impact they claim in the paper. The only novel finding with the help of this model framework is that there might be a new adaptive regime that contributes to the collective yeast glycolysis oscillation. The authors should elaborate on what other novel conclusions can be reached and show how other models may fail to satisfy all of the requirements of published data that theirs does meet.*

Answer 2: Thanks for this constructive criticism. As the referee succinctly summarised in Comment 1, the novelty of our work lies in the quantitative conditions for the transition from quiescent to synchronised oscillations in a cell population, which is formulated in terms of the relevant response functions for the first time. This message was somewhat overshadowed by the emphasis on adaptive response in the previous version, which is more qualitative. Following Referee B’s comment, we have reinstated this key result as the central theme of the paper and organised the more specific findings around it. As Referee A made a similar request in his report, we attach below our answer to his Comment 11:

“In this work, we have established that adaptive cells in a group transform themselves from quiescent to oscillating once certain physical conditions are satisfied. The physical conditions consist of an amplitude relation for the efficiency of intracellular signal relay, and a phase relation which is satisfied by adaptive signal relay and fast signal degradation in the medium. In the revised manuscript, we discussed these issues at length with reference to the two model systems: DQS in dicty and yeast glycolytic oscillations. Our main findings are listed below.

1) *DQS in dicty.* Our quantitative prediction on the oscillation frequency through phase matching agrees well with experiments performed by Thomas Gregor and collaborators (Fig. 6). That DQS requires a sufficiently high cAMP degradation rate in the medium is also borne out by previous experiments. Due to logarithmic sensing of the extracellular cAMP concentration, the adaptive intracellular cAMP relay circuit could support DQS over a broad range of cell densities, as demonstrated by previous experiments and model studies, and is also in agreement with our amplitude relation with $\alpha_2 \sim 1/s$. The actual spatiotemporal dynamics prior to and during aggregation is highly complex and takes several hours to

complete. It is possible that, during this time, cells also change their gene expression and cAMP-related signalling, thus invalidating the assumption of a fixed response spectrum or a fixed set of model parameters. We leave these issues for future study.

2) *Yeast glycolytic oscillations.* Our computational studies show that the ATP auto-catalysis feedback circuit has an adaptive region (for extracellular-intracellular acetaldehyde relay) on the phase diagram spanned by extracellular glucose and acetaldehyde concentrations. The size of this region can be significantly expanded by turning off the glyoxylate shunt in order to strengthen the coupling between ACE signal and ATP homeostasis. For homogeneous cell populations, we predict glycolytic oscillations over a broader range of cell densities due to the adaptive response. Heterogeneities in the cell population and additional ACE clearance mechanisms in the medium can also be considered within our general formalism. Here, more computational work is needed to come up with specific predictions.

3) *Artificial oscillatory units.* As a direct consequence of the amplitude relation, even a single adaptive cell could be induced to oscillate when confined in a sufficiently small volume. This could be considered as a prediction coming out of present work, although conceptually it is quite similar to the hair cell/glass fibre system studied previously in Hudspeth’s lab.”

Apart from the above, we would like to emphasise that, unlike most other models that make specific assumptions about cell dynamics, ours is a general relationship which can be used to check the inner consistency of experimental observations or to propose new targeted experiments to provide the missing information. We continue the discussion on the latter aspect in Answer 4.

Comment 3: *Generality of the model is mostly demonstrated through the three examples buried in the supplement. The generality of the paper could be stressed more if the examples are elaborated in the main text. Indeed, the main text could be expanded from the text of the supplement to help clarify the novel points of the paper. The main text is so short that it is hard to support their claims of novelty.*

Answer 3: Thanks for this great suggestion. We have incorporated the discussion about excitable systems into the main text, under the section Excitable dynamics. We demonstrated that we can predict the cell density and frequency at the onset of oscillations based on the numerically computed linear response functions for the coupled excitable model, thus demonstrating both the generality and predictive power of our approach. The Discussion

section is significantly expanded to include comparison of model studies with experiments. Since most of the pertinent issues are covered, we decided to leave out the analysis of the synthetic circuit.

Comment 4: *There exist many general models that have been used to shed light on potential origins of collective oscillations. This paper could be strengthened by discussing how one could potentially validate or invalidate the theoretical findings of the paper in experimental systems. That is, are there potential knockout experiments or experiments modulating cell density or physical/biochemical environment that would test the claims made here? The authors do specifically say that "In principle, even a single cell (i.e., $N = 1$) could oscillate under the identified mechanism when the coupling constants α_1 and α_2 are sufficiently large, e.g. the cell is confined to a small volume. It would be interesting to perform experiments on cells of different group size and analyze the data using the response function formalism developed here." However, what is interesting about this and what potential validation or breakdown of the model with these experiments would look like and provide that is useful beyond what is already known is not clarified.*

Answer 4: We thank the referee for suggesting us to dig deeper on the experimental side. Despite the generality of the relations we established at the onset of collective oscillations, there are indeed limitations to our theory. The first is the so called "mean-field" assumption that completely ignores possible spatial structures of the extracellular signal. The second is the assumption that cell-to-cell communication is limited to a single extracellular signal while in practice other communication channels may exist. Lastly, DQS may be initiated through a first order transition as seen in certain chemical systems [A.F. Taylor et al., *Science* **323**, 614-617 (2009)]. Some of these complications can be included in an extended modelling framework but due to the lack of experimental information, we leave the discussion to future work.

Given the nature of our results as mentioned in Answer 2, we turned Referee B's questions around in the revised version to emphasise the "transformative" aspect of the physical environment or genetic modifications in inducing collective oscillations in a cell population. As we saw in the example of glycolytic oscillations, cells can be made to adapt better to the extracellular signal when certain reactions are turned on/off. (We are not suggesting that cells actually want to do this.) Changing the activation/feedback timescales τ_a and τ_b

through gene expression or chemical modifications will also have an effect on the activity response function R_a , which in turn affects the group behaviour. The coupling strengths α_1 and α_2 can in principle be tuned by enhancing/reducing secretion of the signalling molecule and by modifying the receptor sensitivity, respectively. Last but not the least, the rate of signal degradation/dilution is known to affect the oscillations. All these issues can now be examined quantitatively using the general relations we established together with data from single-cell measurements. The discussion around Fig. 6 in the revised manuscript illustrates how easily such a procedure can be implemented with FRET and other imaging techniques.

Apart from incorporating the above in various parts of the manuscript, we inserted the following paragraph towards the end of the Discussion section:

These model studies helped to refine and resolve various quantitative issues in the induction of collective oscillations in well-studied systems, and at the same time inspire novel applications built around adaptation-driven signal relay. One promising direction to follow is the development of artificial oscillatory systems with techniques from synthetic biology[61-64]. In analogy with the hair cell/glass fibre setup, one may think of tricking a quorum-sensing cell to oscillate by confining it to a volume small enough to enable positive signal relay.

Below, we answer minor questions from referee B.

Comment 5: *In the paragraph about Fig S5B, the paper notes an opposite trend of frequency shift- coupling relationship between model- predicted and actual data results, but just dropped the statement there. More discussion of this trend would be helpful in understanding why this potentially happens.*

Answer 5: To obtain Fig. S5B, we extended Eq. (4) based on linear response to finite oscillation amplitudes, as discussed in Supplementary III.B. A key assumption for this extension to work is that the higher order harmonic modes of the activity are negligible as compared to the base mode when perturbed by a sinusoidal signal. This is violated in the highly nonlinear FitzHugh-Nagumo model at finite oscillation amplitude, as shown in Fig. S6 in the original version of the manuscript, which we reproduce below.

In the revised manuscript, excitable dynamics is discussed in the Main Text. Given the

FIG. 1: **Power spectra of the signal from simulations of the coupled FHN model.** (A) At $\bar{N} = 1$ which is just above the onset of collective oscillations, the spectrum is dominated by the first harmonic. (B) At $\bar{N} = 2$, spectral weight around the second harmonic becomes significant.

limited space and the large number of topics already included, we defer the prediction of oscillation frequency beyond the critical coupling \bar{N}_o in Fig. 6B to future work.

Comment 6: *In the paragraph discussing routes to collective oscillations in social amoeba, the paper presents two possible routes: adaptive response and single cell limit cycle oscillations. However, experimental work (cited in the paper as refs. 2 and 3) has demonstrated that in the collective natural environment, extracellular cAMP likely never reaches the limit cycle regime.*

Answer 6: We thank the referee for pointing this out. Again, we have moved the example to the Main Text. Due to the limited space and that this is not our main message here, the discussion is removed.

Reviewers' comments:

Reviewer #1 (Remarks to the Author):

I have read the authors' response to my previous comments and the revised manuscript. The revised manuscript did a much better job in presenting the main results with sufficient theoretical details and biological contexts included. I think the revisions have adequately addressed my concerns and criticisms. Therefore, I am happy to recommend publication of the revised manuscript.

Reviewer #2 (Remarks to the Author):

In general the substantially revised manuscript, which includes more biological examples and discussion in the main text, better supports the claims of generality of the proposed model. However, many minor issues remain and could be better explained in the revised text. These include the following:

- In the three-example section (especially the glycolic oscillation example), the paper would be strengthened by further discussions about how the behavior of each individual model mirrors the properties of the predictions and features of the generic model.
- In the discussion section, the paper does further analysis on each example, using the generic model to explain more behaviors such as fold-change detection. However, these are not novel predictions that are unique to this generic model. Are there (novel) predictions that can only or more easily be made out of this generic model but not with other models?
- In weakly nonlinear model (Fig 3D) it's unclear whether the death of collective oscillations with large N is recapitulated by the generic model. For the FHN example (Fig 4C), is there a similar collective oscillation death at large N ?
- For glycolic oscillation example, the signal relay efficiency should be provided in this example and the text could potentially identify a critical value for group oscillation.
- More detail about the work described in Figure 6 could be provided. For example, what parameters are used to achieve a sufficiently high degradation rate to achieve the 6.28 min period?
- The authors should explain the statement starting on line 571 about logarithmic sensing yielding the response amplitude $\alpha^2 \sim 1/s$. The FHN model displays an all-or-none spiking behavior and adding logarithmic sensing doesn't change this.

Reply to Referee B

We thank Referee B for his thoughtful comments and questions, which helped us greatly in improving the clarity of the paper.

Comment 1: *In general the substantially revised manuscript, which includes more biological examples and discussion in the main text, better supports the claims of generality of the proposed model.*

Answer 1: We thank the referee for his positive assessment of the revised manuscript.

Below, we answer the minor issues raised by Referee B.

Comment 2: *In the three-example section (especially the glycolytic oscillation example), the paper would be strengthened by further discussions about how the behavior of each individual model mirrors the properties of the predictions and features of the generic model.*

Answer 2: We thank Referee B for the suggestion. In the 2nd revised manuscript, we inserted texts here and there to better inform the reader how each model study was motivated and performed. Substantial rewriting was made in the section on glycolytic oscillations and in the Discussion section on understanding cAMP oscillations in *Dictyostelium*. We hope our main message now comes across more readily, and the reader, with some work, could link our quantitative procedure to the analysis of collective oscillations in specific biological contexts.

Comment 3: *In the discussion section, the paper does further analysis on each example, using the generic model to explain more behaviors such as fold-change detection. However, these are not novel predictions that are unique to this generic model. Are there (novel) predictions that can only or more easily be made out of this generic model but not with other models?*

Answer 3: Past quantitative experimental and modelling studies on social amoebae have yielded a wealth of information on the intracellular signal relay circuit which is extremely

valuable for testing predictions of our generic model. One technical advance from this work is the direct link between single-cell signal response and collective oscillation frequency as highlighted in Fig. 6. The lower bound for the pulsation interval at 6.28 min can be considered as a novel prediction of our theory. In the 2nd revised manuscript, we rewrote the paragraph following Fig. 6 to make our points more explicit. We hope our work could inspire further experiments that investigate single-cell response and medium properties at different time points of the developmental process, especially those related to possible physiological changes of cells.

Comment 4: *In weakly nonlinear model (Fig 3D) it's unclear whether the death of collective oscillations with large N is recapitulated by the generic model. For the FHN example (Fig 4C), is there a similar collective oscillation death at large N ?*

Answer 4: We thank Referee B again for this excellent question, which prompted us to look into the “reverse DQS”, i.e., viewing the exit from collective oscillations at large N as entry to the oscillatory state from above. Our generic model is designed to describe the emergence/disappearance of collective oscillations through a continuous transition. To address Referee B’s question, a paragraph is added at the end of the section “Necessary conditions for auto-induced collective oscillations” to clarify the applicability of the linear response approach.

In the case of glycolytic oscillations with a small membrane permeability, the transition is continuous and can be captured by our generic model. Please refer to the newly added Fig. S15 and corresponding text in the Supplemental Material for more details.

For our first two examples, i.e., the weakly nonlinear adaptive model and the coupled FHN model, exit from the oscillatory state at high cell densities corresponds to an infinite period bifurcation, accompanied by a discontinuous jump in the oscillation amplitude. This type of sudden death of oscillators has been well studied in the literature but lies outside the main theme of this work.

Comment 5: *For glycolytic oscillation example, the signal relay efficiency should be provided in this example and the text could potentially identify a critical value for group oscillation.*

Answer 5: We thank Referee B for the suggestion. This can indeed be done for the exit

from the oscillatory state on the high density side. Please refer to the newly added Fig. S15 and corresponding text in the Supplemental Material for more details.

Comment 6: *More details about the work described in Figure 6 could be provided. For example, what parameters are used to achieve a sufficiently high degradation rate to achieve the 6.28 min period?*

Answer 6: To answer Referee B’s question, we have modified Figure 6 and added to its caption that “Also shown is the phase shift $-\phi_s$ from Eq. (5) at $\tau_s = 0.35$ min (blue curve), the extracellular signal clearance time to yield an onset oscillation period of about 8 min. Faster signal clearance shortens the oscillation period towards the theoretical lower bound at around 6.28 min.”

Comment 7: *The authors should explain the statement starting on line 571 about logarithmic sensing yielding the response amplitude $\alpha_2 \sim 1/s$. The FHN model displays an all-or-none spiking behavior and adding logarithmic sensing doesn’t change this.*

Answer 7: The term “response amplitude” could indeed be confusing. In the 2nd revised manuscript, we changed it to “detection sensitivity”.

REVIEWERS' COMMENTS:

Reviewer #2 (Remarks to the Author):

We thank the authors for working to clarify the points raised in the previous review. Our scientific concerns are now addressed.